# One-Step In Situ Patternable Reduction of a Ag–rGO Hybrid Using Temporally Shaped Femtosecond Pulses

**DOI:** 10.3390/ma15020563

**Published:** 2022-01-12

**Authors:** Quan Hong, Lan Jiang, Sumei Wang, Ji Huang, Jiaxin Sun, Xin Li, Pei Zuo, Jiangang Yin, Jiangang Lu

**Affiliations:** 1Laser Micro/Nano-Fabrication Laboratory, School of Mechanical Engineering, Beijing Institute of Technology, 5 South Zhongguancun Street, Haidian District, Beijing 100081, China; 159540180@163.com (Q.H.); wangsumei@bit.edu.cn (S.W.); huangji002@126.com (J.H.); sunjiaxin9@gmail.com (J.S.); lixin02@bit.edu.cn (X.L.); zuopei1990@163.com (P.Z.); 2Beijing Institute of Technology Chongqing Innovation Center, Chongqing 401120, China; 3Han’s Laser Technology Industry Group Co., Ltd., 6 Building WanYan Industry Zone, Haoye Road, Fuyong Town, Baoan District, Shenzhen 518103, China; yjg@hanslaser.com (J.Y.); lujg101927@hanslaser.com (J.L.)

**Keywords:** femtosecond laser, photoreduction, hybrid film, Ag–rGO, SERS

## Abstract

In recent years, metallic nanoparticle (NP)–two-dimensional material hybrids have been widely used for photocatalysis and photoreduction. Here, we introduce a femtosecond laser reduction approach that relies on the repetitive ablation of recast layers by usi–ng temporally shaped pulses to achieve the fast fabrication of metallic NP–two-dimensional material hybrids. We selectively deposited silver-reduced graphene oxide (Ag–rGO) hybrids on different substrates under various fabrication conditions. The deposition of the hybrids was attributed to the redistribution of the cooling ejected plume after multiple radiation pulses and the exchange of carriers with ejected plume ions containing activated species such as small carbon clusters and H_2_O. The proposed one-step in situ fabrication method is a competitive fabrication process that eliminates the additive separation process and exhibits morphological controllability. The Ag–rGO hybrids demonstrate considerable potential for chemomolecular and biomolecular detection because the surface-enhanced Raman scattering signal of the enhancement factor reached 4.04 × 10^8^.

## 1. Introduction

Metallic nanoparticle (NP)–two-dimensional material hybrids have been adopted in various fields, such as nanoelectronics [1,2], drug delivery [3], sensors [4,5], catalysis [6,7,8,9], and energy [10]. Such hybrid films not only retain the inherent properties of their component materials but also possess multiple new properties that single-component or multilayer films do not have. For instance, reduced graphene oxide (rGO) and Cu nanowire hybrids can be synthesized to improve the electrical conductivity, oxidation resistance, substrate adhesion, and stability of transparent electrodes [11]. Furthermore, an Ag/rGO@CTFE nanocomposite was presented as a promising photocatalyst for rhodamine B (Rh B) and Evans blue dye degradation under visible light irradiation [12]. Hybrid films have many useful functions and features in different areas.

Common methods of preparing hybrid nanostructures include physical deposition [13,14], chemical reduction [15,16,17], photocatalytic reduction [18], electrochemical deposition, solvothermal methods [17,19], and wave-assisted reduction [20,21,22]. Some of these methods require the use of environmentally unfriendly reducing agents or additional posttreatment processes. Certain other methods without the aforementioned drawbacks cannot precisely control the morphology or stabilize the physical connection and chemical properties of the hybrid. To improve on these methods, simplified processes for fabricating hybrid nanostructures, such as using environmentally friendly reductants [23,24] and employing one-pot [17], one-step [25], and in situ [26] synthesis methods, have been adopted. However, additional posttreatment processes such as filtration, centrifugation, and annealing are still required.

Laser direct writing, a one-step, in situ technique, can solve all these problems and has been used in various areas such as microprocessing and nanoprocessing, multiscale visualization, and surface-enhanced Raman scattering (SERS) applications [27,28,29,30]. Hybrid structures fabricated using laser in situ reduction have been compared with conventional processes. Yu et al. [31] fabricated a TiO_2_–C-based pressure sensor device by using laser radiation and attributed the advantages of the fabrication process to the absence of thermal annealing, which is usually not involved in common, solution-based methods. Rahim et al. [32] presented a Ag–C hybrid for creating flexible, highly conductive traces fabricated using direct laser radiation. When compared with a conventional inkjet printing process, the process was free from the sintering temperature of 200–350 °C. In addition, laser direct writing has been used in SERS applications. Zuo et al. presented a femtosecond reduction process for fabricating Ag–MoS_2_ and Pt–MoS_2_ nanohybrids for SERS and hydrogen evolution reactions. The enhancement factor reached 1.32 × 10^7^, and the detection limit was as low as 10^−11^ M [18]. Teoh et al. provided a straightforward method for microlandscaping Ag NPs on rGO, which enhanced the SERS by up to 16 times [33]. This study demonstrated the patternable reduction of a Ag–rGO hybrid by using temporally shaped femtosecond pulses; the process enhanced SERS by up to 4.04 × 10^8^. During the fabrication process, Ag ions and GO were reduced simultaneously, and hybrid films with controllable morphology were formed through three steps: coating, laser direct writing, and cleaning. Patterned silicon and silica, which are representative of semiconductor and insulator substrates, respectively, were used to prove the wide applicability of the proposed method in different types of materials.

## 2. Materials and Methods

### 2.1. Materials

The GO dispersion (0.5 mg/mL, particles size ranges 0.5–5 μm and stacked layer ranges 1–5) prepared by the common Hummers’ method, was purchased from XFNANO Materials Tech Co., Ltd. (Nanjing, China). Silver nitrate (CAS 7761-88-8), sodium citrate (CAS 68-04-2), and rhodamine 6G (CAS 989-38-8) were obtained from Acros Organics, Fisher Scientific Worldwide (Shanghai) Co., Ltd. (Shanghai, China). Silicon slides (10 mm × 10 mm × 1 mm, 100, double-sided polished) and silica slides (10 mm × 10 mm × 1 mm, double-sided polished) were purchased from Hefei Kejing Materials Tech Co. Ltd. (Hefei, China).

### 2.2. Preparation of the Ag–rGO Precursor

The silver nitrate solution (5 mM) was prepared by adding 4.25 mg of silver nitrate powder to 5 mL of deionized water. The GO dispersion (10 mL, 0.5 mg/mL) was mixed with the silver nitrate solution under sufficient stirring. Subsequently, 3–5 drops (approximately 60 µL per drop) of the mixed solution were homogeneously dispersed on the substrate (Si or SiO_2_) through spin coating at a speed of 500 µm/s for 20 s. The substrate was then dried in a vacuum drying oven (DZF-6034, Shanghai, China) at 50 °C overnight to form the Ag–rGO precursor.

### 2.3. Characterization

Scanning electron microscopy and energy-dispersive X-ray spectroscopy (EDS) images were captured using a microscope (Hitachi SU8220, Tokyo, Japan) operating at an accelerating voltage of 10 kV. The thermogravimetric analysis (TGA) data were recorded on a PerkinElmer Diamond TG-DTA (Waltham, Massachusetts, USA) instrument at a heating rate of 20 °C min^−1^ in N^2^. The SERS signal excited by a 532 nm light source was collected using a Raman spectrometer (Renishaw-InVia-Reflex, Gloucestershire, UK) operating at an exposure time of 10 ms and an excitation power of 0.1% of the laser output power. X-ray photoelectron spectroscopy (XPS) measurements were conducted using a PHI Quantera X-ray photoelectron spectrometer (Chigasaki, Japan) with a monochromatic Al Kα (1486.6 eV) source. The spatial and energy resolutions were 100 μm and 0.48 eV (Ag 3d_5/2_ FHWM), respectively. The TEM results for the Ag NPs were observed using a JEM-2100 (JEOL, Tokyo, Japan) at an acceleration voltage of 80 kV. The samples were prepared by first patterning a 5000 μm × 5000 μm area of the Ag–rGO hybrid on a 10 mm × 10 mm × 1 mm SiO_2_ slide. Subsequently, the slide was immersed in 5 mL of ethanol solution and sonicated for 30 min. Lastly, the droplets of the aforementioned solution were dropped on the carbon-coated electron microscopy grids, and the grids were dried overnight before measurement.

### 2.4. Experimental Setup

The fabrication system consisted of a femtosecond laser (Spitfire Pro-35F1KXP, California, USA, wavelength: 800 nm, repetition rate: 1000 Hz, pulse width: 35 fs), a six-axis Stewart platform (Physik Instrumente M-840.5DG) controlled by a computer, a charge-coupled device camera, a Michelson interferometer (one of the retroreflectors was mounted on a linear transition stage to change the length of one interferometer arm to control the time delay between the divided pulses), and other optical devices for attenuating the power (Appendix A).

### 2.5. Synthesis of the Hybrid Structure on Different Substrates

A silicon or silica substrate with the Ag–rGO precursor layer was mounted on the six-axis Stewart platform, and the Ag–rGO precursor layer was ablated by focusing the laser on the surface. The substrate was then washed with deionized water several times until the entire laser-untreated area was thoroughly cleaned. The residual pattern was dried under ambient conditions.

## 3. Results

### 3.1. Controllable Morphology of the Ag–rGO Hybrid

Because temporally shaped femtosecond pulses can modulate surface structures and photochemical processes by controlling the localized electron density and its distribution in the ablation zone [34,35], the shape, size, and distribution of the Ag–rGO hybrid could be manipulated. We administered overlapping radiation through multiple pulses at different powers with a scanning speed of 200 μm/s and a spot size of 2 μm to obtain a rectangular pattern with varying amounts of Ag–rGO hybrid coating (see Appendix A). The quantities, locations, and shapes of the reductions varied considerably with the fluence. As illustrated in Figure 1, the reduced composite structure changed from an initial coral shape into a granular morphology, and the granules then melted into larger particles. This phenomenon can be attributed to two causes: repetitive fabrication of the recast layer and the different reduction degree of silver. Repeated ablation within a short period in the same area generates a high local temperature, accelerating the melting of the recast layer into either a coralloid-like structure or NPs. The melting of the recast layer can also build a strong connection between the Ag–rGO hybrids and the substrate, which prevents the fabricated structure from being washed away during the cleaning process. At relatively low energy, the ejected plasma could not be sufficiently heated to melt the recast layer into NPs. Therefore, a coralloid-like structure was formed (Figure 1a,b). With suitable energy, this structure then melted into NPs after absorbing sufficient heat (Figure 1c,d). A further increase in the fluence caused the coralloid-like structure to melt and cluster together (Figure 1e,f). Although the shape depended considerably on the fluence, the shape obtained at a specific fluence was fixed, which made the morphology controllable.

The controllable size distribution of the Ag–rGO hybrid when applying a double pulse with different time delays is depicted in Figure 2. The two overlapping pulses with interferometric fringes led to optical intensity fluctuation, which resulted in the discrete distribution of heat. In contrast to the morphology displayed in Figure 2d, the hybrid tended to melt together, and it had a larger size when the two pulses were not separated (Figure 2a,b). However, with an increase in the pulse delay between the two subpulses, the size of the NPs and the density of their distribution decreased. By controlling the distance between the two pulses, different sizes and distributions of Ag–rGO hybrid films were obtained (Figure 2c–f). The distribution density of the Ag–rGO hybrid films reached the minimum value when the delay was 500 fs (Figure 2e). A possible mechanism for this phenomenon was considered to be the energy redistribution and carrier exchange promotion in the second pulse. When GO is laser ablated, it dissociates into small carbon clusters, which subsequently combine to form larger clusters through collision; the size of the GO carbon cluster ions are highly sensitive to the applied laser energy [36]. Separated pulses cause a redistribution of energy and a difference in the excited electron density [34], which then controls the ejection of cluster ions. Therefore, by using these temporally shaped pulses, we can decrease the size of the ejection ions to promote the exchange of carriers.

### 3.2. Characterization of the Ag–rGO Hybrid

During ablation, the Ag–rGO precursor exchanged its carriers and then cooled in the surrounding area to form a recast layer that was assumed to be Ag–rGO [33,37]. To verify the composition, the EDS spectra of the obtained white nanocomposite are presented in Figure 3. Two zones marked by white arrows (see Figure 3a) reveal totally different Ag signals (see Figure 3b,c) indicating that the white coralloid-like structure was composed of silver or silver oxide. Not that the Si and Au signals can be originated from the silicon substrate and the gold spraying process, respectively. 

To further investigate the valence state of the hybrid and the reduction degree of GO, XPS Peakfit software was used [38]; the results are presented in Figure 3d–h. The spectra were fitted using Voigt lineshapes with a mixing ratio M of 0.2 (M = 1.0 represents a pure Lorentzian function) and a Tougaard-type inelastic background [39,40]. XPS Peakfit software was employed [38]. Figure 3d indicates the existence of C, O, and Ag in both the laser-treated and laser-untreated areas. To verify that silver oxide was not contained in the hybrid, binding energy (BE) values of 367.4 eV (AgO) [41], 367.7 eV (Ag_2_O) [42], and 368.3 eV (Ag) [43] were considered when fitting the Ag 3d spectra; no Ag(I) or Ag(II) states were found (Figure 3e), demonstrating the existence of the metallic form of silver. As to the fitting of the C 1 s spectra, different carbon forms such as C sp^2^ as the nonfunctionalized planar carbon, C sp^3^ as the aliphatic nonfunctionalized tetrahedral carbon [44], and oxygen groups such as hydroxyls, carbonyls, and carboxyls should be considered. Moreover, studies have reported that the BE separation values of C sp^3^ and other oxygen groups with respect to C sp^2^ were 0.7 eV (reduced graphene oxide) [40], 1.2–2.5 eV (C–OH), 2.1–3.5 eV (C=O), and 4.0–5.4 eV (C–OOH) [45], respectively. In general, the spectrum can be deconvoluted into six components with BEs of 284.5 eV (sp^2^ carbon in the aromatic ring) [44], 285.2 eV (sp^3^ carbon) [39], 286.2 ± 0.2 eV (hydroxyl C–OH), 287.1 eV (epoxy C–O–C), 288 eV (carbonyl C=O), and 289.2 eV (carboxyl C=O(OH)) [46,47]. The C 1s and O 1s spectra of the Ag–rGO precursor before and after femtosecond reduction and the corresponding atomic contents of C and O are listed in Table 1 and displayed in Figure 3e–h. The decrease of the sp^3^ carbon bond from 34.6% to 19.1% corresponds to the increase of the sp^2^ bond from 29.0% to 39.3% after femtosecond reduction and indicates the conversion of tetrahedral carbon to planar carbon and the partial reduction of GO [39].

The high-resolution transmission electron microscopy results displayed in Figure 4 further reveal the structure of crystalline silver and its orientation. The d-spacing of Ag–rGO was 2.45 Å and close to the value corresponding to Ag (111) (JCPDS no. 89-3722), which suggests the Ag (111) facet on the surface of the Ag NPs. The reduction of the Ag NPs can change the morphology of Ag–rGO hybrid and promote the melting of the recast layer as shown in Appendix A.

### 3.3. Formation Mechanism of the Ag–rGO Hybrid

Studies have examined how H_2_O molecules play an important role in the synthesis of silver nitrate and GO under liquid conditions [33,37,48]. The absorption of H_2_O by GO and the Ag–rGO film ensure the occurrence of an in situ reduction. The TGA results of GO and Ag–rGO (See Appendix A) show a significant mass loss under 100 °C, which can be assigned to the absorption of H_2_O by GO [49,50]. The mechanism of interaction between the Ag–rGO precursor (silver nitrate and graphene oxide) and a laser in a vacuum or under a background gas condition has been investigated in detail. Xu et al. [43,51] discussed the mechanism of the selective metallization of a silver nitrate film on insulators and the localized melting and quasi-welding role of femtosecond direct writing. The reaction for the reduction of a AgNO_3_ film during ablation by a femtosecond laser is assumed to be as follows:(1)$${2\mathrm{AgNO}}_{3}\to {2\mathrm{Ag}+2\mathrm{NO}}_{2}↑{+O}_{2}↑$$

The interaction between GO and a laser has also been investigated. During ablation, the ejected GO plume includes numerous carbon cluster ions. The large carbon cluster dissociates into small carbon clusters, which subsequently combine and cool around the ablation area to form the recasting layer. The ejected plume also includes other activated species such as electrons, atoms, and released gases such as H_2_O, NO_2_, O_2_, and CO_2_ [36,52,53]. 

The ejected plume, which contains carbon cluster ions, plays a key role in the in situ reduction of the Ag–rGO hybrid. A tentative mechanism for this process is illustrated in Figure 5. When the Ag–rGO precursor was ablated by the laser, the electrons and holes of GO were excited, and the critical electron density was quickly exceeded, which resulted in ablation. The ejecta contained carbon cluster ions and silver ions, which exchanged their electrons with the assistance of H_2_O. The in situ reduction of the Ag–rGO hybrid can be represented as follows [33]:(2)$$\mathrm{GO}\overset{\mathrm{hv}}{\to }{\mathrm{GO}(h}^{+}{+e}^{-})$$
(3)$$\mathrm{GO}\overset{\mathrm{Multiple}\;\mathrm{Radiation}}{\to }\mathrm{carbon}\;{\mathrm{clusters}+\mathrm{CO}}_{2}{+H}_{2}O$$
(4)$${4h}^{+}{+H}_{2}O\to O_{2}{+4H}^{+}$$
(5)$${\mathrm{Ag}}^{+}+\mathrm{GO}\left(e^{-}\right)\to \mathrm{GO}+\mathrm{Ag}$$
(6)$${\mathrm{GO}+4e}^{+}{+4H}^{+}\to {\mathrm{rGO}+H}_{2}O$$

### 3.4. Mapping of the Ag–rGO Hybrid and its SERS Application

Flexible or controllable reduction and mapping of Ag–rGO have considerable applications in microdevices and SERS. The high-resolution fabrication of conductive circuits on various substrates under different fabrication conditions is applicable to microdevices. The method of laser direct writing the Ag–rGO composite can be applied in various environments and with various substrates. A microscale “BIT” letter pattern was coated on silica (Figure 6a) by using a far-focus objective lens (50×, NA = 0.5) under ambient environmental conditions. The corresponding EDS mapping images of the pattern are displayed in Figure 6b–d. The results reveal a uniform distribution of C and Ag after reduction by the femtosecond laser (Figure 6b,d). The distribution of O barely changed (Figure 6c); oxygen was partially removed as GO was reduced, and consequently, the amount of oxygen in the laser-treated area was considerably lower than that of silica. The microlandscaping of the Ag–rGO on different substrates under aqueous conditions is illustrated in Appendix A and presented in Appendix A. The successful deposition of silver-reduced graphene oxide (Ag–rGO) hybrids on Si or SiO_2_ under various fabrication conditions proves that this one-step, in situ, patternable reduction method can be used in different substrates.

This method can also be applied for molecular detection; SERS spectroscopy is one of the most popular methods for measuring the sample molecule. The controllable reduction of metallic NPs is critical for obtaining the maximum detection limit because this process enhances excited localized surface plasmon resonance. In this study, the SERS spectra of rhodamine 6G (R6G) were used to verify the application of Ag–rGO. A total of 20 μL of 1 μM R6G solution was dropped on the surface of the hybrid structure, which was dried under ambient environmental conditions. For comparison, 20 μL of 10 mM R6G solution was dropped on the surface of the non-laser-treated area. The SERS signal was collected using a 532 nm laser. For the quantitative analysis of the enhanced Raman signal, the estimation of the enhancement (G) factor is essential. This can be calculated from the following equation [54]:(7)$$\mathrm{EF}=\frac{I_{\mathrm{SERS}}}{I_{\mathrm{ref}}}\times \frac{N_{\mathrm{ref}}}{N_{\mathrm{SERS}}}\times \frac{P_{\mathrm{ref}}}{P_{\mathrm{SERS}}}$$
where I_SERS_ is the measured SERS intensity for the probe molecules on the nanoparticle surface, I_ref_ is the measured intensity of normal Raman scattering from the substrate, N_SERS_ is the concentration of the probe molecules used on the Ag–rGO hybrid sample, N_ref_ is the concentration of the probe molecules used on the substrate, P_SERS_ is the power used for probing the Raman signal on the concentration of the probe molecules, and P_ref_ is the power used for probing the Raman signal on the substrate. Here, the enhancement factor was calculated for the R6G Raman spectra with a 532 nm laser excitation. The prominent band at 1649 cm^−1^ (aromatic C-C stretching vibration [55] was chosen for calculating the EF. Because neat R6G is highly fluorescent, 0.01 M R6G aqueous solution was analyzed for the substrate Raman spectra. The concentration of R6G adsorbed on the rGO–Ag thin film was 1 × 10^−6^ M. With the assumption that the R6G molecules spread homogeneously on the sample, the ratio between the R6G molecules was predicted to be in direct proportion to the ratio between the concentrations. To make the Raman signal of the substrate evident, the probe power was enhanced to 100 times relative to that of the Ag–rGO hybrid. Figure 7 indicates that the enhancement factor varied with the fluence and the delay between the two subpulses. As shown in Figure 7a,b, the maximum enhancement factor reached was EF = 28699/71 × 100 × 10^4^ = 4.04 × 10^8^ when the fluence was 0.708 J/cm^2^ and the delay was 1 ps. The fluctuation of the enhancement factor was mainly caused by the different shapes and distributions of metallic NPs.

## 4. Conclusions

In summary, femtosecond laser direct writing of the Ag–rGO hybrid was demonstrated to be a method unlimited by the substrate material or processing environment. Patterning of the Ag–rGO hybrid on the silicon and silica substrates was displayed under ambient and aqueous conditions. By using temporally shaped femtosecond pulses, the fabrication and transfer of the hybrid were considerably simplified. The controllability of the proposed approach was demonstrated by examining the morphology with changes in the fluence and the delay between the two subpulses. The thermal effect accumulation of multiple pulses and the synergistic effect between GO and Ag NPs were assumed to be the two main reasons for the controllability of the reduction. The characterization results further verified that silver NPs and graphene oxide were simultaneously reduced. Moreover, the Ag–rGO hybrid was applied to an SERS application with good Raman signal enhancement, indicating excellent potential for photoreduction and photocatalysis.

## Figures and Tables

**Figure 1 materials-15-00563-f001:**
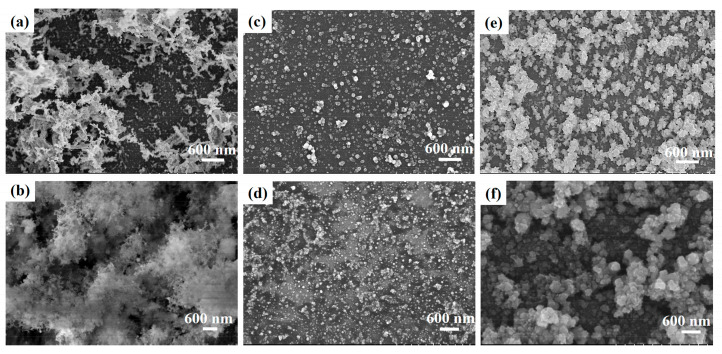
Micromorphology of the Ag–rGO hybrid on SiO_2_ treated with a femtosecond laser at the following fluences: (**a**) 0.035 J/cm^2^, (**b**) 0.071 J/cm^2^, (**c**) 0.178 J/cm^2^, (**d**) 0.354 J/cm^2^, (**e**) 0.708 J/cm^2^, and (**f**) 1.288 J/cm^2^.

**Figure 2 materials-15-00563-f002:**
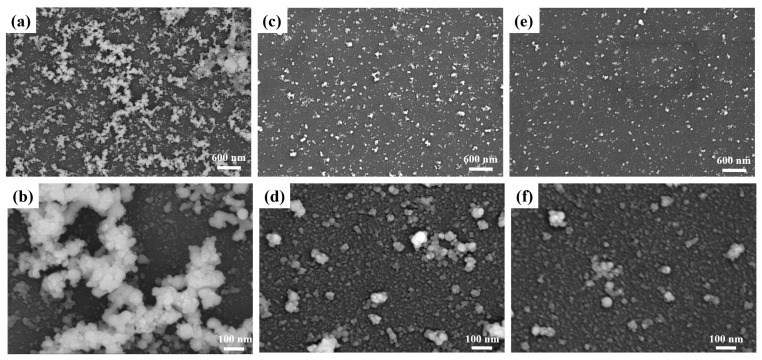
Micromorphology of the Ag–rGO hybrid on SiO_2_ treated using a femtosecond laser with different time delays, a fluence of 0.354 J/cm^2^, and pulse widths of (**a**,**b**) 0 fs, (**c**,**d**) 300 fs, and (**e**,**f**) 500 fs.

**Figure 3 materials-15-00563-f003:**
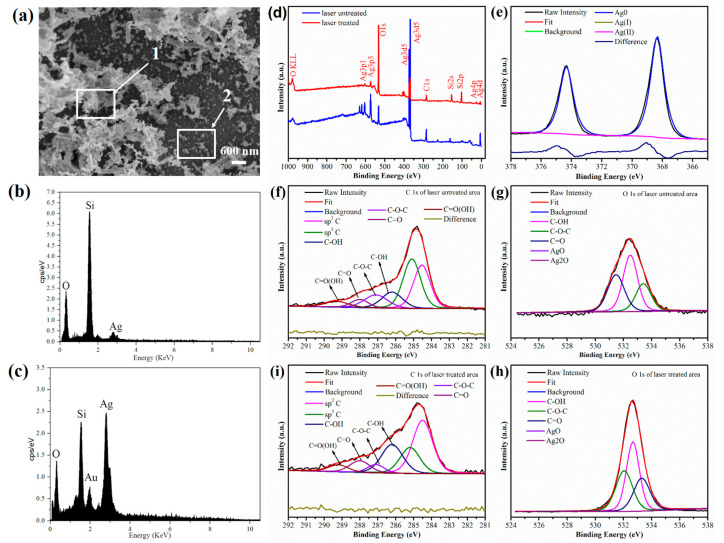
Characterization of the Ag–rGO film. (**a**) Morphology of the coralloid-like Ag–rGO hybrid when the fluence was 0.035 J/cm^2^, (**b**) EDS results for the zone indicated by arrow 2, (**c**) EDS results for the zone indicated by arrow 1. XPS Peakfit software results for the Ag–rGO precursor before and after femtosecond laser direct writing: (**d**) complete spectra of the Ag–rGO precursor, (**e**) Ag 3d spectra of the Ag–rGO precursor after laser direct writing. (**f**,**g**) C 1s and O 1s spectra of the Ag–rGO precursor before laser direct writing, (**h,i**) C 1s and O 1s spectra of the Ag–rGO precursor after laser direct writing.

**Figure 4 materials-15-00563-f004:**
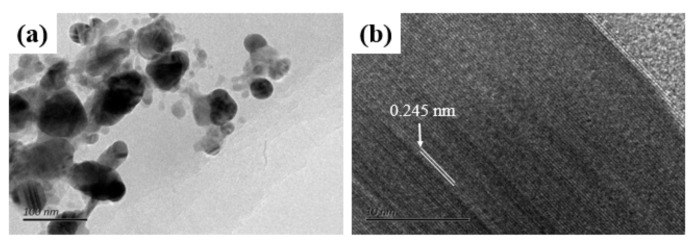
High-resolution transmission electron microscopy images of the Ag–rGO hybrid; (**a**) scar bar = 100 nm, (**b**) scar bar = 10 nm.

**Figure 5 materials-15-00563-f005:**
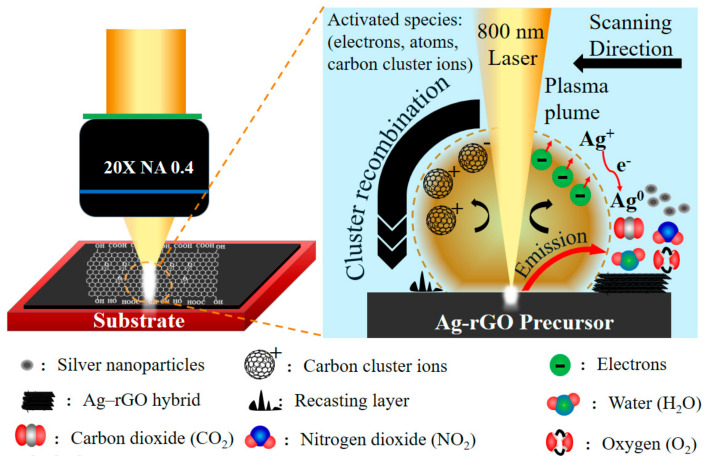
Formation of the Ag–rGO hybrid during ablation by a femtosecond laser.

**Figure 6 materials-15-00563-f006:**
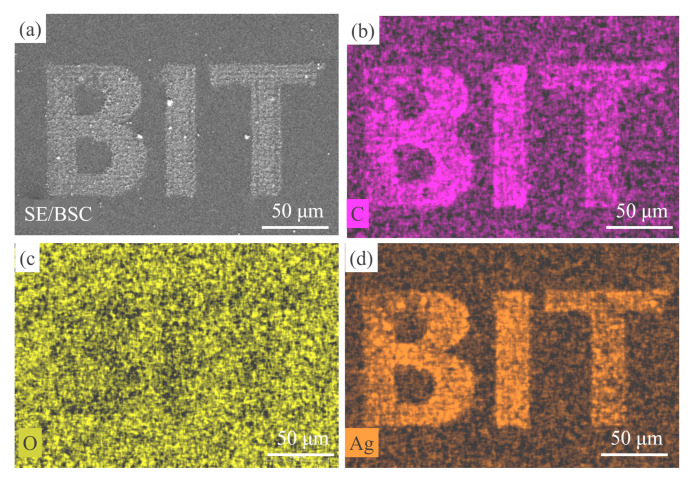
Mapping of the Ag–rGO hybrid on the silicon substrate and the elemental distribution of the hybrid obtained through energy-dispersive X-ray spectroscopy: (**a**) signal collecting area, (**b**) C, (**c**) O, and Ag(**d**).

**Figure 7 materials-15-00563-f007:**
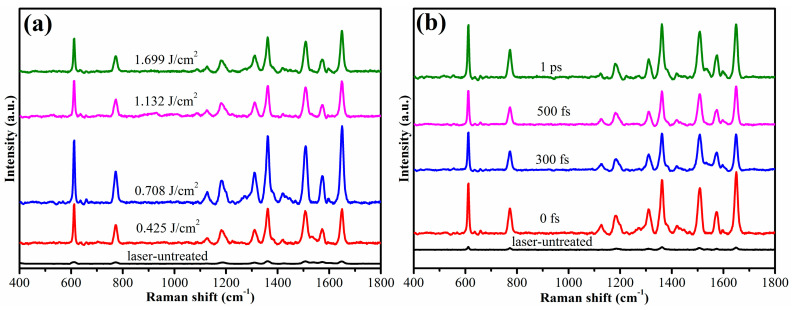
Raman spectra of R6G when the power was varied from 0.178 to 1.399 J/cm^2^ (**a**) and when the time delay was varied from 0 to 1 ps at a fixed power of 0.708 J/cm^2^ (**b**).The concentration of R6G used in the laser-untreated area was 10^−2^ M, and the concentration used in other areas was 10^−6^ M).

**Table 1 materials-15-00563-t001:** Atomic content of the C and O chemical groups before and after laser ablation.

**Before Femtosecond Ablation**	**C 1s group content (at. %), (** **BE (eV))**					
	C sp^2^ (284.5)	C sp^3^ (285.2)	C-OH (286.2)	C-O-C (287.1)	C=O (288.0)	C-OOH (289.2)
	29.0	34.6	13.8	12.1	4.5	6.0
	**O 1s group content (at. %), (** **BE (eV))**					
	C-OH (532.6)		C-O-C (533.3)		in carboxyl group C=O (531.9)	
	35.1		23.7		35.7	
**After Femtosecond Ablation**	**C 1s group content (at. %), (** **BE (eV))**					
	C sp^2^ (284.5)	C sp^3^ (285.2)	C-OH (286.4)	C-O-C (287.1)	C=O (288.0)	C-OOH (289.2)
	39.3	19.1	22.7	5.2	8.7	5.0
	**O 1s group content (at. %), (** **BE (eV))**					
	C-OH (532.6)		C-O-C (533.3)		in carboxyl group C=O (531.9)	
	42.5		25.5		32.0	

## Data Availability

The data presented in this study are available on request from the corresponding author. At the time the project was carried out, there was no obligation to make the data publicly available.

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
