# Peer review of "One-Step In Situ Patternable Reduction of a Ag–rGO Hybrid Using Temporally Shaped Femtosecond Pulses"

_materials, 2022, doi:10.3390/ma15020563_

Round 1
Reviewer 1 Report
I read the manuscript "One-Step in Situ Patternable Reduction of a Ag−rGO Hybrid Using Temporally Shaped Femtosecond Pulses" with interest and do not find much problem. I would like to suggest the following points before recommending publication.
- The raw energy-dispersive X-ray spectroscopy spectrum should be shown as a figure in the main manuscript. I would like see the atomic ratio of C, O, and Ag.
- Fig. 3 (c) should be labelled as "C 1s of Laser untreated area" and (e) should be as "C 1s of Laser treated area".
- The Raman spectra should also show the assignments of the oscillation modes.
Reviewer 2 Report
1) Materials and methods. More detailed characterization of GO precursor is needed (particle size, mean number of layers, C/O ratio etc.)
2) “ several drops of the mixed solution …”
- The amount of precursor per 1 cm2 of substrate and the layer thickness of the dried precursor should be given
3) “The substrate was then dried in the vacuum drying oven …”
- The amount of residual water in the precursor layer should be characterized by TG-MS or other method. It is strongly recommended to add to this curve a TG curve of the dried GO dispersion without AgNO3 for comparison.
4) p.3, “This phenomenon can be attributed to … the different reduction degree of silver.”
p.5, “no Ag(I) or Ag(II) states were found, demonstrating the existence of the metallic form of silver”
- Which of these statements is true?
5) Fig.4. It would be reasonable to add similar micrographs of the initial GO and/or rGO without Ag for comparison in order to identify the effect of AgNO3 on the morphology of hybrid particles.
6) p.7, “the in situ reduction of a Ag−rGO hybrid without the participation of H2O has remained unclear”
“The ejecta contained carbon cluster ions and silver ions, which exchanged their electrons with the assistance of H2O.”
- You have to correct these statements using the information on the amount of residual H2O in the precursor obtained from TG data (see comment (3)).
7) p.9, “The maximum reach enhancement factor was EF = 4.04 × 10(8) when the fluence was 0.708 J/cm2 and the delay was 1 ps.”
- You have to explain the reasons of non-monotonous variation of EF with the fluence, when it first grows until 0.708 J/cm2 then drops down. It seems reasonable to add the TEM micrographs of Ag-rGO hybrids obtained at the different fluences; they could be helpful for this discussion.
8) “were homogeneously dispersed on the substrate (Si or SiO2) through spin coating”; “Patterning of the Ag−rGO hybrid on the silicon and silica substrates” etc. etc.
- You have to explain why different substrates were used. Have you observed the effect of substrate material on the results of these experiments?
